# 6′-Sialyllactose Alleviates Muscle Fatigue through Reduced Blood Lactate Level after Treadmill Exercise in Mice

**DOI:** 10.3390/nu16172957

**Published:** 2024-09-03

**Authors:** Eun-Jung Park, Li-La Kim, Jie-Oh Lee, Hay-Young Lee, Yong-An Kim, Hiroe Go

**Affiliations:** 1GeneChem Inc., Daejeon 34025, Republic of Korea; sksgy@genechem.co.kr (E.-J.P.); lilack8899@gmail.com (L.-L.K.); 2Department of Life Sciences, Pohang University of Science and Technology (POSTECH), Pohang 37673, Republic of Korea; jieoh@postech.ac.kr; 3POSTECH Biotech Center, Pohang University of Science and Technology (POSTECH), Pohang 37673, Republic of Korea; hayyoung@postech.ac.kr

**Keywords:** 6′-sialyllactose, muscle fiber type, anti-fatigue, lactate, oxidative phosphorylation protein complexes

## Abstract

6′-Sialyllactose (6′-SL), found in human breast milk, exhibits anti-inflammatory, immune function-enhancing, brain development-promoting, and gut health-improving effects. However, its effects on muscle fatigue remain unknown. Here, we aimed to investigate the effects of 6′-SL on blood lactate level, muscle fiber type, and oxidative phosphorylation protein complexes (OXPHOS) in muscle after exercise using C57BL/6J male mice. C57BL/6J mice were randomly assigned to control or 100 mg/kg 6′-SL. After 12 weeks of 6′-SL administration, the mice were made to perform treadmill exercise; their blood lactate and glucose levels were measured at the basal level (rest) and 0, 5, and 10 min after treadmill exercise. Results showed that 6′-SL treatment in C57BL/6J mice significantly reduced blood lactate level and improved blood glucose level. Moreover, 6′-SL increased the expression of slow-myosin heavy chain (MHC) and OXPHOS in gastrocnemius muscle. In addition, 6′-SL treatment for 12 weeks did not affect food intake, serum biomarkers of tissue injury, and lipid profiles compared with those of the controls. These findings indicate that non-toxic 6′-SL suppressed muscle fatigue during exercise by promoting protein expression of muscle fibers, especially slow-twitch muscle fibers characterized by abundant OXPHOS complexes and decreased blood lactate level. This study suggests that 6′-SL holds promise as a nutritional supplement in exercise and clinical settings, subject to further validation.

## 1. Introduction

During exercise with limited oxygen supply, lactate production increases through anaerobic metabolism in muscles; thus, lactate serves as a marker of fatigue [1,2]. Lactate clearance post-exercise is essential for recovery and performance enhancement [1]. Optimizing lactate clearance can improve athletic performance and overall fitness [3]. Recovery strategies may involve increasing mitochondrial density, enhancing lactate transport, and maintaining glycolytic capacity in muscle fibers [4,5,6]. Dietary interventions and exercise training can induce these adaptations [7,8]. However, long-term safety and potential adverse effects, such as gastrointestinal distress and drug interactions, need consideration [9,10]. Moreover, some supplements lack sufficient evidence regarding their athletic performance enhancement [11]. Addressing these concerns requires safe and accessible treatments.

Skeletal muscle has two main fiber types: slow-twitch and fast-twitch. Slow-twitch fibers are fatigue-resistant, mitochondria-rich, and mainly use aerobic metabolism, specifically the oxidative phosphorylation (OXPHOS) pathway, for ATP production [12,13]. This pathway includes enzymes and protein complexes such as NADH dehydrogenase, succinate ubiquinone oxidoreductase, ubiquinol cytochrome c oxidoreductase, cytochrome c oxidoreductase, and ATP synthase. They facilitate electron transport and ATP synthesis within the mitochondria [14]. Slow-twitch fibers are essential for endurance activities such as endurance exercise [15]. Fast-twitch fibers mainly rely on anaerobic metabolism for quick ATP production, crucial for power and speed but can lead to fatigue due to by-products such as lactate [15,16].

6′-Sialyllactose (6′-SL), a human milk oligosaccharide (HMO) found abundantly in breast milk, is involved in several bioactive functions such as enhancing immune function, maintaining gut microbiota balance, and promoting brain development; it also serves as a soluble decoy receptor for viruses, bacteria, and protozoan parasites [17,18]. Moreover, 6′-SL has been shown to improve symptoms of UDP-*N*-acetylglucosamine 2-epimerase/*N*-acetylmannosamine kinase myopathy in vivo (mouse) and in clinical settings [19,20]. 6’-SL improves endurance exercise performance in *Caenorhabditis elegans* by increasing clustered morphology of mitochondria, decreasing fatty acid levels, and increasing glycogen use during exercise [21]. Increased mitochondrial function can produce more ATP efficiently using the OXPHOS pathway, thus relying less on anaerobic glycolysis, which is the primary source of lactate production [22].

Based on the above findings, 6’-SL may also be beneficial against muscle fatigue during exercise by affecting muscle characteristics and mitochondrial function. To test this hypothesis, in the present study, we aimed to investigate the effects of 6’-SL on lactate production, a marker of fatigue during exercise, muscle fiber type, and OXPHOS complex protein expression in mouse. 

## 2. Materials and Methods

### 2.1. Animal and Treatments

All animal experiments were performed according to the guidelines of the Pohang Technopark Animal Ethics Committee (ABCC 2022009; Pohang, Republic of Korea). This study was performed with 4-week-old male C57BL/6J mice, weighing 19–22 g, from Daehan Biolink (Seoul, Republic of Korea). The mice were individually housed in a controlled environment (22–23 °C, 12-:12-h light/dark cycle) and fed a standard diet; they had free access to water. Animal health was monitored by animal activity, panting, fur condition, body weight, body weight gain, and food intake. Animal activity, panting, and fur condition were monitored once daily, five days per week. Body weight, body weight gain, and food intake were measured weekly. After three weeks of acclimatization to the experimental facility, the mice (*n* = 20) were randomly divided into two groups, control (*n* = 10) and 6′-SL groups (*n* = 10), based on body weight measurements taken by investigators. Once a day, five days per week, the control group mice were orally administered water, and 6′-SL group mice were administered 100 mg/kg 6′-SL for twelve weeks. Weight gain was calculated as the difference in body weight between weeks. Physical characteristics (body weight, body weight gain, and food intake), serum biomarker measurements, blood lactate and blood glucose measurements, and western blotting were assessed in this study. 6′-SL was produced using the one pot reaction system invented by GeneChem, Inc. (Daejeon, Republic of Korea) and dissolved in water. Briefly, substrates and enzymes (cytidylate kinase (CMK), polyphospate kinase (PPK), CMP-NeuAc synthetase (NEU), *N*-acetyl-D-glucosamine-2-epimerase (NANE), NeuAc aldolase (NAN), and α2,6-sialyltranferase (α2,6STN)) were mixed and reacted in a one-pot reactor. These enzymes were obtained from a strain of beta-D-galactosidase deficient *E. coli* BW25113 that originated from non-pathogenic *E. coli* K-12. All procedures were approved by the institutional ethics committee for the care and use of animals.

### 2.2. Serum Analysis

After humanely euthanizing the mice, blood samples were collected. The samples were then centrifuged at 6000 rpm for 10 min at 4 °C to obtain serum, which was then stored at −80 °C until further analysis. The serum was used for the analysis of alanine aminotransferase (ALT), aspartate aminotransferase (AST), alkaline phosphatase (ALP), T-bilirubin, albumin (ALB), total protein, blood urea nitrogen (BUN), creatinine, low-density lipoprotein (LDL), high-density lipoprotein (HDL), triglyceride (TG), and total cholesterol (TC) levels using an autoanalyzer (TBA-40FR; Toshiba, Tochigi, Japan). 

### 2.3. Treadmill Exercise

At 12 weeks after 6′-SL administration, the mice were made to perform treadmill exercise (Mouse Treadmill, Ugo basile, Gemonio, Italy). The mice were first acclimated to a speed of 16.67 m/min for 3 min, and then, the speed was increased by 1.67 m/min every 3 min; the total running time was set to 20 min.

### 2.4. Blood Lactate and Glucose Measurements

At 12 weeks after 6′-SL administration, blood samples were collected from the tail tip at the basal level (rest) and 0, 5, and 10 min after the 20-min treadmill exercise session to assess the blood glucose and lactate levels using a handheld glucometer (ACCU-check, Seoul, Republic of Korea) and a lactate test meter (Lactate Pro 2, Kyoto, Japan), respectively.

### 2.5. Western Blotting

After humanely euthanizing the mice, gastrocnemius (GAS) muscles were harvested from the mice. The GAS muscle was frozen in liquid nitrogen for protein extraction. Total protein from the GAS muscle was extracted via homogenization using T-PER^TM^ Tissue Protein Extraction Reagent (78510; Thermo Fisher, Rockford, IL, USA) and centrifuged at 15,000 rpm for 15 min at 4 °C. The supernatant was added to 5× SDS sample buffer and boiled for 5 min at 100 °C or 1 h at 37 °C. The supernatant was then collected, and protein levels were measured using the Quick Start^TM^ Bradford Protein Assay Kit (5000202; Bio-Rad Laboratories, Hercules, CA, USA). Equal amounts of samples (20 μg) were loaded onto 4–20% Mini-PROTEAN^®^ TGX^TM^ Precast Protein Gels (4561096; Bio-Rad Laboratories, Hercules, CA, USA) and resolved. Thereafter, the resolved proteins were transferred onto polyvinylidene difluoride membranes using the Trans-Blot Turbo Transfer System (25 V for 7 min) (1704156; Bio-Rad Laboratories). The membranes were blocked with casein blocking buffer (IBS-BC052; iNtRON Biotechnology, Boston, MA, USA) for 30 min at room temperature (25 °C ± 1) and incubated overnight at 4 °C with primary antibodies: anti-slow MHC (1:250, M8421; Sigma-Aldrich, St. Louis, MO, USA), anti-fast MHC (1:1000, ab91506; Abcam, Cambridge, UK), anti-OXPHOS (1:1000, ab110413; Abcam), and anti-α-tubulin (1:1000, sc-5286; Santa Cruz Biotechnology, Dallas, TX, USA). The following day, the membranes were washed with Tris-buffered saline containing 0.1% Tween (TBST) three times for 10 min and incubated with HRP-conjugated goat anti-rabbit or anti-mouse IgG secondary antibodies (1:10,000, LF-SA8001 and LF-SA8002; Abfrontier, Seoul, Republic of Korea) at room temperature (25 °C ± 1) for 2 h. After washing three times with TBST, the protein bands were detected with ECL reagent (1705061; Bio-Rad Laboratories) using a ChemiDoc XRS Imaging System (Universal Hood II, Bio-Rad Laboratories) with the Image Lab 6.0 software (Bio-Rad Laboratories).

### 2.6. Statistical Analysis

Data are presented as mean ± SD. Two-tailed unpaired *t*-test or two-way analysis of variance followed by Tukey’s multiple comparisons test were used to evaluate the differences between the groups. The significance level was set at *p* < 0.05. For all statistical analyses, GraphPad Prism 10.1.0 (GraphPad Software, Inc., San Diego, CA, USA) was used. The sample size for each experiment was selected to calculate significant differences. The number of animals was adjusted between the control and 6′-SL treatment groups. For physical characteristics, serum biomarker measurements, and blood lactate and blood glucose measurements, data from eight mice were included for significance calculation. For Western blotting, data from three mice were included for significance calculation. Animals were excluded if they died prematurely or if tissue samples were lost during autopsy.

## 3. Results

### 3.1. Effects of 6′-SL on Physiological Characteristics

To confirm the effects of 6′-SL on physiological characteristics in mice, body weight, weight gain, food intake, and serum biomarker levels were measured. 6′-SL increased body weight at 7, 9, 10, and 12 weeks, and weight gain was observed at 1 and 7 weeks (Figure 1A,B). Food intake remained unchanged for 12 weeks (Figure 1C). In the 6’-SL group, the serum BUN level decreased, but the liver and kidney biomarker levels, as well as lipid profiles, showed no significant differences compared with those in the control group (Table 1). These results indicate that 6′-SL is safe to administer for up to 12 weeks without causing any organ damage.

### 3.2. 6′-SL Decreases the Level of Fatigue-Related Factors during Exercise

We confirmed the effects of 6′-SL on blood lactate level, which is a useful biological marker of fatigue in the exercising muscles. 

The lactate level of the 6′-SL treated group (3.0 ± 0.35 mmol/L) was lower than that of the control group (4.3 ± 0.35 mmol/L) at rest. Immediately after the treadmill exercise (0 min), lactate levels in the 6′-SL treated group (5.5 ± 0.76 mmol/L) decreased significantly compared to that in the control group (10.8 ± 3.47 mmol/L) and remained around the same level (6.3 ± 0.96–7.3 ± 2.06 mmol/L) for 10 min (Figure 2A). On the contrary, the control group showed an increase in the lactate level immediately after the treadmill exercise (0 min) (10.8 ± 3.47 mmol/L), and the level remained high for 10 min (9.4 ± 2.49–10.9 ± 3.13 mmol/L) (Figure 2A). Moreover, the level of Area Under the Curve (AUC) also showed decreased blood lactate in the 6′-SL treated group (87.47) compared to the control group (142.7) (Figure 2A).

The glucose level was lower in the 6′-SL group compared with that in the control group at rest; it increased at 5 min and significantly increased at 10 min after the treadmill exercise (*p* > 0.01) (Figure 2B). However, the level of AUC was similar between two groups (Control: 2200, 6′-SL: 2235) (Figure 2B).

Furthermore, the serum BUN level, a fatigue-related factor, decreased in the 6′-SL group compared with that in the control group (control: 23.25 ± 3.24 mg/dL, 6′-SL: 19.75 ± 2.92 mg/dL) (Table 1). These results suggest that 6′-SL decreases blood lactate level and promotes blood glucose production during exercise.

### 3.3. 6′-SL Increases Energy Metabolism in the GAS Muscle

To examine the effect of 6’-SL on MHC isoform, protein expression of slow and fast MHC was measured in the GAS muscle. 6′-SL notably elevated slow MHC protein expression (*p* < 0.05) but did not affect fast MHC expression (Figure 3A). Slow MHC is rich in mitochondria and efficiently produces ATP through aerobic metabolism. Here, 6’-SL increased the expression of slow MHC, and thus efficient energy source metabolism through an aerobic mechanism was expected to occur. 

To confirm this, we examined the changes in protein expression of the OXPHOS complex, which is involved in aerobic metabolism. As shown in Figure 3B, the expression of complexes I (*p* < 0.01), III (*p* < 0.05), IV (*p* < 0.05), and V (*p* < 0.01) increased in the 6′-SL group compared with that in the control group. These results suggest that 6′-SL increases energy metabolism by increasing slow MHC expression in the GAS muscles.

## 4. Discussion

The present study demonstrated that 6′-SL increased protein expression of slow MHC and OXPHOS complexes in the GAS muscle and blood glucose level, and decreased blood lactate level. Furthermore, 6′-SL significantly increased the expression of slow MHC, which is found in slow-twitch muscle fibers.

Muscle fiber type is associated with exercise performance capacity. The ratio of slow-twitch to fast-twitch muscle fibers plays an important role in contractile events, energy homeostasis, muscle force, and fatigue [23,24]. Slow-twitch muscle fibers are crucial for slow but continuous ATP synthesis and contractile events [24,25]. Therefore, muscles with increased amounts of slow-twitch muscle fibers are more resistant to fatigue and perform better during prolonged exercise [24,25]. In contrast, fast-twitch muscle fibers facilitate rapid ATP production, support muscle speed and force, and are less fatigue-resistant [24,25].

Slow-twitch muscle fibers contain a large number of mitochondria, allowing them to obtain ATP via oxidative metabolism [26]. Moreover, the number of slow-twitch muscle fibers correlates with muscle fitness [26]. For example, a reduced number of slow-twitch muscle fibers diminishes energy metabolism, consequently leading to several diseases such as metabolic disorders, heart failure, and muscular dystrophy [26]. Improvement in muscle function by exercise training is induced by an increase in the proportion of slow-twitch muscle fibers [26]. Moreover, endurance exercise is known to increase respiratory capacity and the ratio of slow-twitch muscle fibers [27,28]. Myricetin enhances anti-fatigue ability by switching fast- to slow-fiber type [29]. 

These reports support the idea that 6′-SL-indcued protein expression of slow MHC will enhance anti-fatigue in muscle (Figure 3A). However, the precise mechanism of 6’-SL-induced slow-MHC protein expression is not yet clear. It will be necessary to investigate the muscle fiber switching through evaluating the expression of each muscle fiber type (type I, type IIa, type IIb, and type IIx) using western blotting and immunofluorescence, as well as the expression of myogenic regulatory factors, myostatin, paired box family, and forkhead box transcription factors, which are known to control myofiber development and differentiation [30].

In muscle cells, OXPHOS complex protein expression is upregulated during exercise to increase energy supply, and this process involves OXPHOS-mediated lactate oxidation [31,32]. Moreover, mitochondrial dysfunction is known to result in excess fatigue and other symptoms associated with chronic disease and muscle deterioration [33]. Clinical trials have shown that oral dietary supplements such as L-carnitine, alpha-lipoic acid, coenzyme Q10, reduced nicotinamide adenine dinucleotide (NADH), and membrane phospholipids can restore mitochondrial function and reduce fatigue [33]. *Platycodon grandiflorum*-derived saponins enhance exercise performance by increasing the expression of OXPHOS-III and -IV complex proteins [34]. 

6′-SL increased protein expression of OXPHOS complexes in the GAS muscle in our study (Figure 3B). Further study will be needed to evaluate molecular mechanisms involved in effects of 6’-SL on OXPHOS complex proteins. The relevance of protein kinases such as cAMP-dependent protein kinase A, PTEN-induced putative kinase 1, cyclin-dependent kinase 11, SRC, and 5′-AMP-activated protein kinase, which are known to regulate the OXPHOS system, could improve the quality of this study [35].

Several factors cause muscle weakness and fatigue. For example, aggregation of metabolites, such as lactic acid, ammonia, BUN, and inorganic phosphorus, that induce intracellular acidosis has been shown to cause muscle fatigue [36]. Efforts to eliminate these metabolic products and to recover energy production can reverse fatigue. For example, *Moringa oleifera* extracts increase swimming endurance by reducing the accumulation of lactate, BUN, and ammonia [37]. *Glossogyne tenuifolia* extracts increase grip strength and endurance swimming time by decreasing lactate level [36]. 

Endurance exercise demands high energy levels and consumes glucose from tissue glycogen in the liver and muscle [38]; reduced blood glucose levels have been shown to cause fatigue [39]. Moreover, for contraction of working muscles, glucose is used as a source of energy to produce ATP, which is the immediate fuel source of the body. 

In our study, 6′-SL decreased blood lactate levels and increased blood glucose levels after the treadmill exercise (Figure 2). Therefore, we speculate that 6′-SL may enhance exercise performance by decreasing lactate and BUN levels and enhancing glucose supply in the blood. Further studies to investigate how 6’-SL regulate lactate metabolism during exercise would be interesting. For example, changes in glycolysis (glycolysis hexokinase), tricarboxylic acid cycle enzymes (citrate synthase, succinate dehydrogenase, malate dehydrogenase, pyruvate dehydrogenase), oxygen consumption, and ATP production may be worth investigating. Moreover, measuring the level of lactate in muscle and liver and assessing the relevance of Cori cycle will more precisely explain how blood lactate levels decline.

Plasma fatty acids and intramuscular triglycerides are energy sources used during exercise [40]. To verify this, we assessed the effects of 6′-SL on the serum levels of LDL, HDL, TG, and TC. The findings showed no alterations in the levels of these parameters, suggesting that fats were not used as an energy source in this study (Table 1). 

To confirm the safety of 6′-SL, we assessed its effects on liver and kidney tissue damage. Assessment of ALT, AST, ALP, T-bilirubin, ALB, total protein, and creatine, the key markers of tissue damage, revealed that 6′-SL did not change their levels, suggesting no organ damage during exercise performance (Table 1). After 6′-SL administration for 12 weeks, increased body weight was observed (Figure 1). As this increase was within the range of normal average body weight change, we suggest that it does not have considerable significance. However, it cannot be ruled out that the weight of muscles or bones related to exercise increased with 6′-SL administration for 12 weeks. However, food intake and body weight gain were not significantly different between the groups after 12 weeks (Figure 1). These results indicate that the mice grew at a steady rate during treatment. The safety of 6′-SL in healthy adults has also been reported [41].

This study has several limitations. First, a single dose of 6′-SL (100 mg/kg) was administered for a single duration (12 weeks) in this study. It is necessary to optimize the effect of 6′-SL by conducting experiments under various conditions, such as the concentration at which there is no effect, concentration at which there is the best effect, and minimum period for which the effect appears in the future. Second, only one type of exercise was tested to measure blood lactate and glucose levels. Treadmill running is physically demanding and is an effective exercise to burn energy and induce fatigue; therefore, this type of exercise was chosen in this study to measure blood lactate and blood glucose levels after inducing fatigue. However, in the future, another type of exercise, such as swimming, should be conducted to obtain more detailed results on the effectiveness of 6’-SL in reducing lactate levels. Moreover, exhausted exercise performance such as treadmill running, swimming, and rotarod test should be performed to confirm whether the reduction in blood lactate level is indeed associated with improved exercise performance in the future. Third, the efficacy of 6’-SL in reducing muscle fatigue is more robust when compared to other nutritional supplements known to be effective in removing lactate from the blood, such as omega-3, CoQ10, magnesium, and acetyl-L-carnitine. To support these findings and develop 6’-SL as a dietary supplement to alleviate muscle fatigue, exploring clinical relevance will be required. 

## 5. Conclusions

In conclusion, this study demonstrated that 6′-SL increased muscle content, especially slow MHC, which contains abundant mitochondria. Therefore, 6′-SL increased the expression of OXPHOS complexes to produce energy using glucose as an energy source, resulting in decreased lactate levels during treadmill exercise. Moreover, serum biomarker levels were not changed. These results suggest that 6’-SL is beneficial and safe in reducing muscle fatigue during exercise, indicating future utility in athletic and clinical settings after further validation. 

## Figures and Tables

**Figure 1 nutrients-16-02957-f001:**
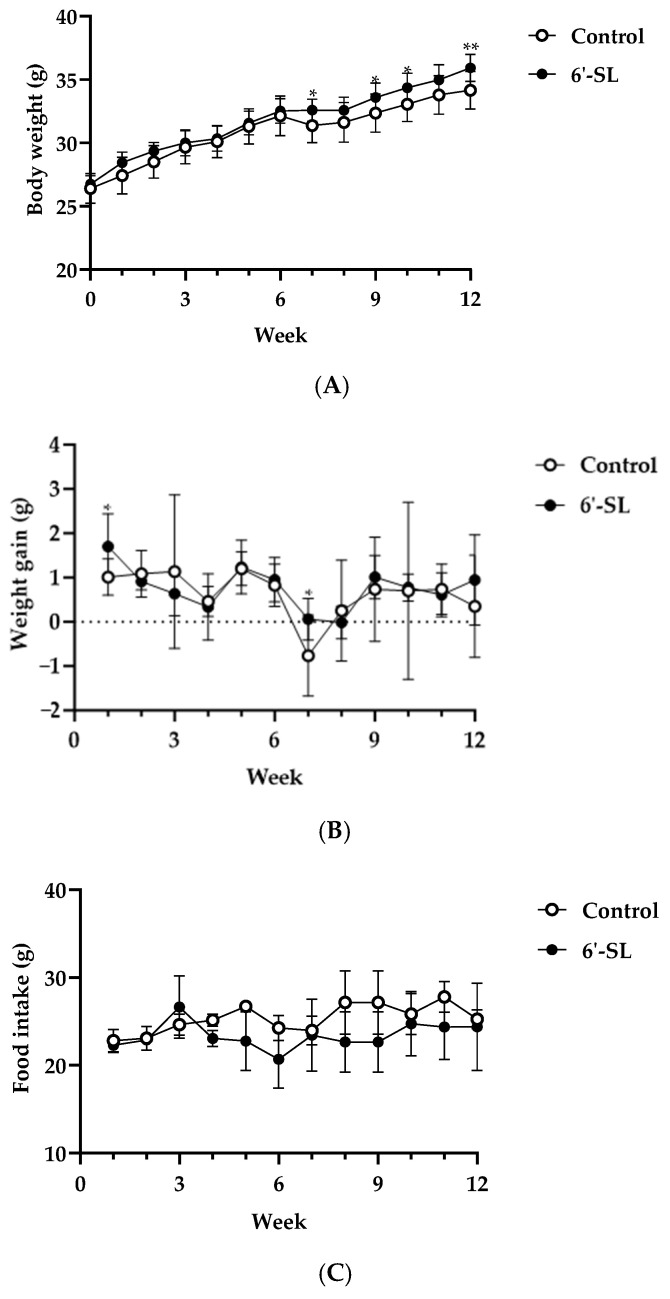
Effect of 6′-SL on physical characteristics. (**A**) Body weight, (**B**) weight gain, and (**C**) food intake were measured at weekly intervals during the experiment period. Data are presented as mean ± SD for each group (*n* = 8). A two-way analysis of variance followed by Tukey’s multiple comparisons test was used; * *p* < 0.05, ** *p* < 0.01 vs. control.

**Figure 2 nutrients-16-02957-f002:**
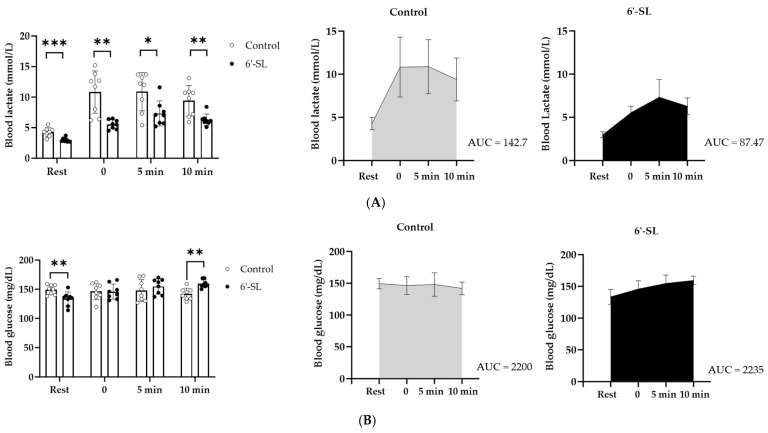
Effect of 6′-SL on lactate and glucose levels in the blood. (**A**) Blood lactate and (**B**) glucose levels were measured at rest and 0, 5, and 10 min after the treadmill exercise. The total AUC of blood lactate (**A**) and glucose (**B**) were calculated. Data are presented as mean ± SD for each group (*n* = 8). A two-way analysis of variance followed by Tukey’s multiple comparisons test was used; * *p* < 0.05, ** *p* < 0.01, *** *p* < 0.001 vs. control.

**Figure 3 nutrients-16-02957-f003:**
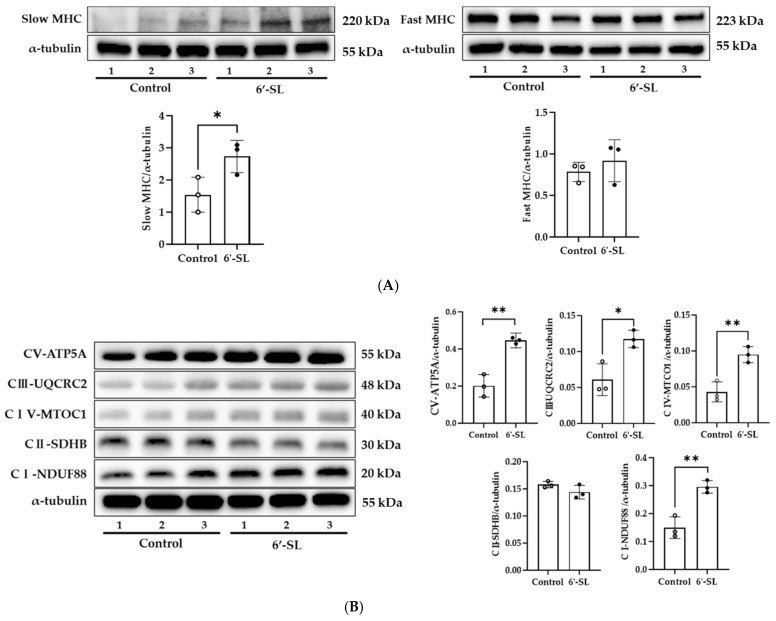
Effect of 6′-SL on muscle differentiation marker and OXPHOS complexes in the GAS muscle. Protein expression levels of (**A**) slow and fast MHC, and (**B**) OXPHOS complexes of the GAS muscle were measured using Western blotting. NADH dehydrogenase (CI-NDUF88), succinate ubiquinone oxidoreductase (CII-SDHB), ubiquinol cytochrome c oxidoreductase (CIII-UQCRC2), cytochrome c oxidoreductase (CIV-MTOC1), and ATP synthase (CV-ATP5A). The relative band intensity of each protein was normalized to that of α-tubulin, and the quantified data are shown in the bar graphs. White dots mean control groups and black dots mean 6′-SL groups. Data are presented as mean ± SD for each group (*n* = 3). A two-tailed unpaired *t*-test was used; * *p* < 0.05, ** *p* < 0.01 vs. control.

**Table 1 nutrients-16-02957-t001:** Levels of serum biomarkers of liver and kidney injury and fatigue and lipid profiles.

	Control (*n* = 8)	6′-SL (*n* = 8)	*p*-Value
Liver injury marker			
ALT (U/L)	22.25 ± 12.31	14.25 ± 5.50	0.1155
AST (U/L)	113.75 ± 25.76	104.75 ± 25.66	0.4953
ALP (U/L)	41.75 ± 2.05	41.75 ± 5.60	>0.9999
T-bilirubin (mg/dL)	0.06 ± 0.01	0.05 ± 0.01	0.2663
ALB (d/dL)	2.78 ± 0.19	2.65 ± 0.21	0.2298
Total protein (g/dL)	3.91 ± 0.24	3.70 ± 0.21	0.0798
Kidney injury marker			
Creatinine (mg/dL)	0.41 ± 0.08	0.37 ± 0.03	0.1905
Fatigue marker			
BUN (mg/dL)	23.25 ± 3.24	19.75 ± 2.92 *	0.0395
Lipid profiles			
LDL (mg/dL)	12.38 ± 5.24	9.75 ± 1.67	0.1981
HDL (mg/dL)	70.13 ± 9.60	67.75 ± 6.54	0.5722
TG (mg/dL)	72.25 ± 19.25	63.50 ± 15.41	0.3325
TC (mg/dL)	85.13 ± 13.59	79.75 ± 8.10	0.3529

Data are presented as mean ± SD for each group (*n* = 8). A two-tailed unpaired *t*-test was used; * *p* < 0.05 vs. control. Abbreviations: 6′-SL, 6′-sialyllactose; ALT, alanine aminotransferase; AST, aspartate aminotransferase; ALP, alkaline phosphatase; T-bilirubin, total-bilirubin; ALB, albumin; BUN, blood urea nitrogen; LDL, Low density lipoprotein; HDL, high density lipoprotein; TG, triglyceride; TC, total cholesterol; SD, standard deviation.

## Data Availability

The datasets generated during and/or analyzed during the current study are available from the corresponding author on reasonable request.

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
