# Peer review of "6′-Sialyllactose Alleviates Muscle Fatigue through Reduced Blood Lactate Level after Treadmill Exercise in Mice"

_nutrients, 2024, doi:10.3390/nu16172957_

Round 1

Reviewer 1 Report

Comments and Suggestions for Authors

This manuscript demonstrates that long- term 6′-Sialyllactose supplementations can alleviate post- exercise muscle fatigue by increasing slow- twitch muscle fibers expression, blood glucose and reducing lactate levels. It’s a topic of interest to the researchers in the areas of sports nutrition but the paper needs some significant improvements. The detailed comments are as follows:

1.     Title: (1) The content of the title is not fully in line with the theme of the manuscript, the word “Suppresses” in the title is too absolutist, it’s recommended to be adjusted, and it is recommended to add the causative factor of muscle fatigue (exercise) in it. (2) The title should contain the time of supplement and the time of exercise.

2.     Abstract: (1) The concept of lactate clearance is mentioned in the abstract, but it is not shown in the results, so it’s suggested to modify this part of the abstract or calculate the lactate clearance by the corresponding formula and show it in the result section. (2) When the exercise was done during 12 weeks, or after 12 weeks’ supplements? What kind of the exercise? (3) This study was an observation by animal experiment, its prospective utility in human, like athletic and clinical settings, is an inference application, which is not suitable to write down directly.

3.     Methods: (1) A description of the number of mice per group and the total number is missing, and sample size calculations should be performed for this to qualify as statistically significant.

(2) It is recommended that more detail on the content of the 12-week post-supplementation exercise be provided in this section.

(3) What kind of the exercise on the treadmill? Intensity and time? Why choose it?

4.     Results: (1) In lines 145 and 150, the results section should provide only an objective description of the results of the experiment, with as little interpretation or conclusion of the results as possible (this section should be placed in the discussion).

(2) Line 124, the food intake remained unchanged for 12 weeks, which is doubt for readers. The mice from four-week-old, adding three weeks of acclimatization, and 12-week intervention, were 19 weeks old at last. During the high growth period for rodents, food intake should increase according to the body weight (Fig1 A). There is also a doubt on the weight gain (Fig1 B), the gain of each group was -1~2g, which is different with the value (for example, 35g-26g=9g, between 12week and 0 week).

(3) Line 140, the title only talked blood lactate, but the data of blood glucose and BUN are shown in the following. The figure in here shows the comparism between control and 6’SL at each time point, it is suggested to draw diagram of curves of lactate and glucose, and calculate the AUC and the lactate clearance rate. (4) It is insufficient for proteins expression analysis by n=3.

5.     Discussion: (1) Compared to the control group, the experimental group itself resulted in additional calorie intake, and this contributed to the significant increase in body weight, it is suggested that an explanation of the weight-related indicators could be added to the discussion section. (2) The kind and duration of exercise would influence the energy consumption, then it is suggested that distinguish between different types of exercise when discussing. (3) It would be helpful if the limitations of this study are added more explicitly.

Author Response

For Brief Report

Response to Reviewer 1 Comments

1. Summary

2. Questions for General Evaluation

Reviewer’s Evaluation

Response and Revisions

Does the introduction provide sufficient background and include all relevant references?

Must be improved

Is the research design appropriate?

Must be improved

Are the methods adequately described?

Must be improved

Are the results clearly presented?

Must be improved

Are the conclusions supported by the results?

Can be improved

3. Point-by-point response to Comments and Suggestions for Authors

Comments 1: Title: (1) The content of the title is not fully in line with the theme of the manuscript, the word “Suppresses” in the title is too absolutist, it’s recommended to be adjusted, and it is recommended to add the causative factor of muscle fatigue (exercise) in it. (2) The title should contain the time of supplement and the time of exercise.

Response 1: Thank you for pointing this out. We agree with this comment.

(1) The content of the title is not fully in line with the theme of the manuscript, the word “Suppresses” in the title is too absolutist, it’s recommended to be adjusted, and it is recommended to add the causative factor of muscle fatigue (exercise) in it.

→ We have changed title “6´-Sialyllactose suppresses muscle fatigue through reduced blood lactate” to “6′-Sialyllactose Alleviates Muscle Fatigue through Reduced Blood Lactate Level after Treadmill Exercise in Mice”.

(Page 1, Title, Line 2)

(2) The title should contain the time of supplement and the time of exercise.

→ We want to emphasize that 6´-SL itself decreased muscle fatigue in this study. We think

that several durations of supplement should be tested in the future.

→ After long time exercise, blood lactate is saturated, and no differences are determined between groups. Therefore, we measured it after 20 min of treadmill exercise. However, important thing here is that the lactate was reduced, so I don't think it's necessary to include information such as the exercise duration in the title.

Comments 2: Abstract: (1) The concept of lactate clearance is mentioned in the abstract, but it is not shown in the results, so it’s suggested to modify this part of the abstract or calculate the lactate clearance by the corresponding formula and show it in the result section. (2) When the exercise was done during 12 weeks, or after 12 weeks’ supplements? What kind of the exercise? (3) This study was an observation by animal experiment, its prospective utility in human, like athletic and clinical settings, is an inference application, which is not suitable to write down directly.

Response 2: Thank you for pointing this out. We agree with this comment.

(1) The concept of lactate clearance is mentioned in the abstract, but it is not shown in the results, so it’s suggested to modify this part of the abstract or calculate the lactate clearance by the corresponding formula and show it in the result section

→ We have modified the word "increased lactate clearance" has been changed to "decreased blood lactate.".

 (Page 1, Abstract, Line 25-26)

(2) When the exercise was done during 12 weeks, or after 12 weeks’ supplements? What kind of the exercise?

→ After 12 weeks of 6′-SL administration, the mice were made to perform treadmill exercise; their serum lactate and glucose levels were measured at the basal level (rest) and 0, 5, and 10 min after treadmill exercise.

(Page 1, Abstract, Line 17-19)

(3) This study was an observation by animal experiment, its prospective utility in human, like athletic and clinical settings, is an inference application, which is not suitable to write down directly.

→ We have modified the sentence “indicating its prospective utility in athletic and clinical

settings after further validation” to “This study suggest that 6′-SL holds promise as a

nutritional supplement in exercise and clinical settings, subject to further validation.

(Page 1, Abstract, Line 26-27)

Comments 3: Methods: (1) A description of the number of mice per group and the total number is missing, and sample size calculations should be performed for this to qualify as statistically significant. (2) It is recommended that more detail on the content of the 12-week post-supplementation exercise be provided in this section. (3) What kind of the exercise on the treadmill? Intensity and time? Why choose it?

Response 3: Thank you for pointing this out. We agree with this comment.

(1) A description of the number of mice per group and the total number is missing, and sample size calculations should be performed for this to qualify as statistically significant.

→ We have added the number of mice per group and the total number.

(Page 2. Section 2.1., Line 77-78)

→ The number of mice in each group and each experiment was selected to be at least three, and significant differences were calculated using GraphPad Prism 10.1.0.

We have explained how the sample size was decided as follows:

For physical characteristics, serum biomarker measurements, serum lactate and blood glucose measurements, eight mice were included for significance calculation. For Western blotting, three mice were included for significance calculation.

(Page 3, Section 2.6., Line 131-134)  

(2) It is recommended that more detail on the content of the 12-week post-supplementation exercise be provided in this section.

→ We explained the methods of blood lactate and glucose measurement as follows:

2.3. Treadmill Exercise

At 12 weeks after 6′-SL administration, the mice were made to perform treadmill exercise. The mice were first acclimated to a speed of 16.67 m/min for 3 min, and then, the speed was increased by 1.67 m/min every 3 min; the total running time was set to 20 min.

2.4. Blood Lactate and Glucose Measurements

At 12 weeks after 6′-SL administration, blood samples were collected from the tail tip at

the basal level (rest) and 0, 5, and 10 min after the 20-min treadmill exercise session to assess the blood glucose and lactate levels using a handheld glucometer (ACCU-check, Seoul, South Korea) and a lactate test meter (Lactate Pro 2, Kyoto, Japan), respectively.

(Page 2-3, Section 2.3. & Section 2.4., Line 93-101)

(3) What kind of the exercise on the treadmill?

Running was conducted on treadmill.  

Intensity and time?

→ Intensity and time were added to the method section as follows:    

2.3. Treadmill Exercise

At 12 weeks after 6′-SL administration, the mice were made to perform treadmill exercise. The mice were first acclimated to a speed of 16.67 m/min for 3 min, and then, the speed was increased by 1.67 m/min every 3 min; the total running time was set to 20 min.

(Page 2-3, Section 2.3., Line 93-96)

Why choose it?

→ In this study, we measured blood lactate as a fatigue substance. Running on a treadmill

is physically demanding and an effective way to burn energy and induce fatigue.

Therefore, we selected the running treadmill. However, we should conduct another type

of exercise, like swimming to get more detailed results on the effectiveness of 6′-SL in the

future. We have added the explanation in discussion as follows:

Treadmill running is physically demanding and is an effective exercise to burn energy and induce fatigue; therefore, this type of exercise was chosen in this study to measure blood lactate and blood glucose levels after inducing fatigue. However, in the future, another type of exercise, such as swimming, should be conducted to obtain more detailed results on the effectiveness of 6'-SL in reducing lactate level. Moreover, exhausted exercise performance such as treadmill running, swimming, and rotarod test should be performed to confirm whether the reduction in blood lactate level is indeed associated with improved exercise performance in the future.

  (Page 8, Section 4, Line 249-256)

Comments 4: Results: (1) In lines 145 and 150, the results section should provide only an objective description of the results of the experiment, with as little interpretation or conclusion of the results as possible (this section should be placed in the discussion).

(2) Line 124, the food intake remained unchanged for 12 weeks, which is doubt for readers. The mice from four-week-old, adding three weeks of acclimatization, and 12-week intervention, were 19 weeks old at last. During the high growth period for rodents, food intake should increase according to the body weight (Fig1 A). There is also a doubt on the weight gain (Fig1 B), the gain of each group was -1~2g, which is different with the value (for example, 35g-26g=9g, between 12week and 0 week).

(3) Line 140, the title only talked blood lactate, but the data of blood glucose and BUN are shown in the following. The figure in here shows the comparism between control and 6’SL at each time point, it is suggested to draw diagram of curves of lactate and glucose, and calculate the AUC and the lactate clearance rate. (4) It is insufficient for proteins expression analysis by n=3

Response 4: Thank you for pointing this out. We agree with this comment.

(1) In lines 145 and 150, the results section should provide only an objective description of the results of the experiment, with as little interpretation or conclusion of the results as possible (this section should be placed in the discussion).

→ We have transferred the sentence “For contraction in working muscles, glucose is used as a source of energy to produce ATP, which is the immediate fuel source of the body” to discussion section.

(Page 8, Section 4, Line 243-244)

→ We have modified the sentence “These results suggest that 6′-SL decreases fatigue and produces an energy source for the muscles during exercise.” to “These results suggest that 6′-SL decreases blood lactate level and promotes blood glucose production during exercise."

  (Page 5, Section 3.2., Line 173-174)

(2) Line 124, the food intake remained unchanged for 12 weeks, which is doubt for readers. The mice from four-week-old, adding three weeks of acclimatization, and 12-week intervention, were 19 weeks old at last. During the high growth period for rodents, food intake should increase according to the body weight (Fig1 A). There is also a doubt on the weight gain (Fig1 B), the gain of each group was -1~2g, which is different with the value (for example, 35g-26g=9g, between 12week and 0 week).

→ Animal health is monitored by food intake and body weight. An excessive increase or decrease in food intake indicates something is wrong with the mouse's health. The average body weights of C57BL/6J mice from 7 to 19 weeks have been reported to be 23.6±1.5 g and 33.7±2.8 g, respectively (Body Weight Information for C57BL/6J (000664) from the Jackson Laboratory). The average weight results in this study are within this reported range, which means that the weight and food intake values ​​are correct.

→ Weight Gain indicates how much weight mice have gained since the previous week. Therefore, this result means body weight of mice increased steady rate during experiments. We have added an explanation to make it easier to understand in method section as follows:

  Weight gain was calculated as the difference in body weight between weeks.

 (Page 2, section 2.1., Line 81-82)

(3) Line 140, the title only talked blood lactate, but the data of blood glucose and BUN are shown in the following. The figure in here shows the comparism between control and 6’SL at each time point, it is suggested to draw diagram of curves of lactate and glucose, and calculate the AUC and the lactate clearance rate

→ We have modified the title “6′-SL Decreases Blood Lactate during Exercise” to “6′-SL Decreases the Level of Fatigue-Related Factors during Exercise”

(Page 5, Section 3.2., Line 155)

→ The AUC for lactate and glucose were calculated and explained in the results section as follows:

  - Moreover, the level of Area Under the Curve (AUC) also showed the decreased blood lactate in 6′-SL treated group (70.44) compared to control group (112.5).

(Page 5, Section 3.2., Line 165-166)

- The glucose level was low in the 6′-SL group compared with that in the control group

at rest; it increased at 5 min and significantly increased at 10 min after the treadmill exercise (P > 0.01) (Figure 2B). However, the level of AUC was similar between two groups (Control: 1609, 6′-SL: 1677).

(Page 5, Section 3.2., Line 167-170)

→ The important thing in this study is that the rate of lactate production was reduced in the 6′-SL group compared with the control group. Furthermore, the 10 minutes after the treadmill was too short to see any significant changes in lactate clearance, and in fact there was no significant difference in clearance. Therefore, we have not added the lactate clearance rate in the manuscript.

(4) It is insufficient for proteins expression analysis by n=3

→ The number of mice was selected to be at least three to calculate significant differences.

This number is sufficient to analysis significant differences between two groups using

GraphPad Prism 10.1.0.

Comments 5: Discussion: (1) Compared to the control group, the experimental group itself resulted in additional calorie intake, and this contributed to the significant increase in body weight, it is suggested that an explanation of the weight-related indicators could be added to the discussion section. (2) The kind and duration of exercise would influence the energy consumption, then it is suggested that distinguish between different types of exercise when discussing. (3) It would be helpful if the limitations of this study are added more explicitly.

Response 5: Thank you for pointing this out. We agree with this comment.

(1) Compared to the control group, the experimental group itself resulted in additional calorie intake, and this contributed to the significant increase in body weight, it is suggested that an explanation of the weight-related indicators could be added to the discussion section.

→ We have added an explanation of weight-related indicators to the discussion as follows:

In this study, after 6′-SL administration for 12 weeks, increased body weight was observed. As this increase was within the rage of normal average body weight change, we suggest that it does not have considerable significance. However, it cannot be ruled out that the weight of muscles or bones related to exercise increased with 6′-SL administration for 12 weeks.

. (Page 9, Section 4, Line 268-272)

(2) The kind and duration of exercise would influence the energy consumption, then it is suggested that distinguish between different types of exercise when discussing.

→ We have added an explanation of different types of exercise to the discussion as follows:

Treadmill running is physically demanding and is an effective exercise to burn energy and induce fatigue; therefore, this type of exercise was chosen in this study to measure blood lactate and blood glucose levels after inducing fatigue. However, in the future, another type of exercise, such as swimming, should be conducted to obtain more detailed results on the effectiveness of 6'-SL in reducing lactate level. Moreover, exhausted exercise performance such as treadmill running, swimming, and rotarod test should be performed to confirm whether the reduction in blood lactate level is indeed associated with improved exercise performance in the future.

(Page 8, Section 4, line 249-256)

(3) It would be helpful if the limitations of this study are added more explicitly.

→ We have added the limitation of this study in the discussion as follows:

Further studies to investigate how 6'-SL is related to lactate clearance in the body would be interesting. For example, changes in glycolysis (glycolysis hexokinase), tricarboxylic acid cycle enzymes (citrate synthase, succinate dehydrogenase, malate dehydrogenase, pyruvate dehydrogenase), oxygen consumption, ATP production, and the Cori cycle may be worth investing.

Measuring the level of lactate in muscles and the liver will allow more precise explanation of the mechanism by which blood lactate level is reduced. The expression of each muscle fiber type (type I, type IIa, type IIb, and type IIx) needs to be examined using western blotting and immunofluorescence to examine muscle fiber switching during 6'-SL administration.

The limitations of this study are that a single dose of 6′-SL (100 mg/kg) was administer for a single duration (12 weeks). It is necessary to optimize the effect of 6′-SL by conducting experiments under various conditions, such as the concentration at which there is no effect, concentration at which there is the best effect, and minimum period for which the effect appears in the future.

(Page 9, Section 4, line 273-287)

Lastly, I would like to thank you for your time and effort in helping me to improve the quality of this paper. Thank you for your comments, and I hope that my revisions and modifications will meet your expectations.

Reviewer 2 Report

Comments and Suggestions for Authors

Dear Authors,

Thank you for submitting your manuscript investigating the effects of 6′-sialyllactose (6′-SL) on muscle fatigue, specifically its influence on blood lactate concentration during exercise, muscle fiber type, and oxidative phosphorylation protein complexes (OXPHOS) in C57BL/6J mice. Her study aims to demonstrate whether 6′-SL treatment can reduce serum lactate levels and improve the expression of proteins associated with muscle endurance, such as slow-twitch muscle fibers.

Your research addresses an important question regarding muscle performance and fatigue, with significant implications for athletic performance and potential clinical applications in the management of muscle fatigue. However, I have carefully reviewed your study and have identified several key areas that require significant revision. Below are the detailed items that need to be addressed to improve the quality and clarity of your manuscript.

MAIN POINTS
Lack of information on the activity regime of the mice: The manuscript lacks detailed information on the activity regime of the mice when they are not undergoing experimental procedures. Please provide comprehensive details on the daily activity levels of the mice, including whether they were kept inactive or at rest in their cages. This information is critical as it can significantly affect the results and interpretation of the study.

Insufficient details on the treadmill exercise protocol: There is insufficient information on the duration and intensity of the treadmill exercise performed by the rats. Please describe the treadmill exercise protocol in detail. This should include frequency, duration, intensity, and any progressive adjustments in difficulty. It is also important to clarify whether the treadmill exercise was performed only at the time of evaluation or as part of a regular exercise regimen.

Insufficient statistical analysis: The statistical analysis presented needs to be corrected. Studies with two groups and at least two assessment times should be analyzed using a two-way ANOVA with a repeated measures factor (in this case, time). Especially when using GraphPad Prism. Revise the statistical methods to include a two-way repeated measures ANOVA. This approach accounts for within-subject variability and provides a more accurate analysis of the data. Also, what normality test was used?

Text Clarity: The text presented between lines 64 and 66 suggests that 6′-SL may improve health and quality of life. Please review this section and provide a clear rationale for this suggestion as it was not evaluated in the reported study. In addition, ensure that appropriate references are provided to support this claim.

ADDITIONAL WEAKNESSES
Complex sentences and redundancy: Some sentences are overly complex and there are instances of redundant phrases and repetition of ideas. Please simplify complex sentences by breaking them into shorter, more digestible sentences. Eliminate redundant phrases to make the manuscript more concise.

Transitions between sections: Transitions between sections and paragraphs could be smoother. Add transitional phrases or sentences to move the reader more seamlessly through the manuscript. This will improve overall flow and coherence.

Specificity in Methods and Results: Certain sections, especially the Methods and Results, lack specific details and explanations. Please provide more specific details and explanations in these sections to ensure complete clarity and understanding. This will improve the reproducibility of your study.

MINOR ISSUES
Figures: Revise figures to use the same font as the text. Make sure figures are centered within the text of the manuscript and do not take up the entire page.

Formatting Issues:
I. There is an extra parenthesis after the word "biotechnology" on line 109. Please remove the extra parenthesis to correct the formatting.
II. The letters in line 253 are larger than the text. Please adjust the font size to match the rest of the manuscript text.

English language improvements: While the manuscript is generally clear and free of major grammatical errors, further improvements can be made to enhance readability. Make sure that all sentences are well constructed and that the flow of ideas is logical and coherent. Also, continue to use technical terms and concepts correctly and appropriately. Finally, simplify complex sentences, eliminate redundancies, and improve transitions to make the manuscript more accessible and effective for readers.

Addressing these issues, large and small, will greatly improve the rigor, clarity, and overall quality of your manuscript. Thank you for your attention to these details, and I look forward to seeing the improved version of your paper.

Comments on the Quality of English Language

The manuscript is generally clear and free of major grammatical errors. Sentences are well constructed and the flow of ideas is logical and coherent. Technical terms and concepts are used correctly and appropriately, contributing to the overall readability and comprehension of the manuscript.

However, some sentences are overly complex and could be simplified to improve readability. Breaking long sentences into shorter, more digestible ones can help readers follow the argument more easily. In addition, there are instances of redundant phrases and repetition of ideas. Streamlining the text to avoid repetition would make the manuscript more concise.

Transitions between sections and paragraphs could be smoother. Adding transitional phrases or sentences would help guide the reader more seamlessly through the manuscript. In some sections, particularly the methods and results, more specific details and explanations would ensure complete clarity and understanding.

Specific suggestions include simplifying complex sentences, eliminating redundancies, and improving transitions. By addressing these areas, you can improve the quality of the English in your manuscript, making it more accessible and effective for your readers.

Author Response

For Brief Report

Response to Reviewer 2 Comments

1. Summary

2. Questions for General Evaluation

Reviewer’s Evaluation

Response and Revisions

Does the introduction provide sufficient background and include all relevant references?

Yes

Is the research design appropriate?

Can be improved

Are the methods adequately described?

Must be improved

Are the results clearly presented?

Can be improved

Are the conclusions supported by the results?

Yes

3. Point-by-point response to Comments and Suggestions for Authors

Comments 1: Lack of information on the activity regime of the mice: The manuscript lacks detailed information on the activity regime of the mice when they are not undergoing experimental procedures. Please provide comprehensive details on the daily activity levels of the mice, including whether they were kept inactive or at rest in their cages. This information is critical as it can significantly affect the results and interpretation of the study.

Response 1: Thank you for pointing this out. We agree with this comment.

We checked animal health by body weight, body weight gains, and food intake. We have not found that severe change of these factors during acclimatization and experiments. We have added the information on the activity regime in method section as follows:

Animal health was monitored once daily, 5 days per week by body weight, body weight gain, and food intake.

(Page 2, Section 2.1., Line 75-77)

Food intake and body weight gains showed no significant differences after 12 weeks. These results indicate that the mice grew at a steady rate during treatment.

(Page 9, Section 4, line 264-266)

Comments 2: Insufficient details on the treadmill exercise protocol: There is insufficient information on the duration and intensity of the treadmill exercise performed by the rats. Please describe the treadmill exercise protocol in detail. This should include frequency, duration, intensity, and any progressive adjustments in difficulty. It is also important to clarify whether the treadmill exercise was performed only at the time of evaluation or as part of a regular exercise regimen.

Response 2: Thank you for pointing this out. We agree with this comment.

We have added the detailed information on the treadmill exercise protocol in method

section as follows:

2.3. Treadmill Exercise

At 12 weeks after 6′-SL administration, the mice were made to perform treadmill exercise. The mice were first acclimated to a speed of 16.67 m/min for 3 min, and then, the speed was increased by 1.67 m/min every 3 min; the total running time was set to 20 min.

2.4. Blood Lactate and Glucose Measurements

At 12 weeks after 6′-SL administration, blood samples were collected from the tail tip at

the basal level (rest) and 0, 5, and 10 min after the 20-min treadmill exercise session to assess

the blood glucose and lactate levels using a handheld glucometer (ACCU-check, Seoul, South Korea) and a lactate test meter (Lactate Pro 2, Kyoto, Japan), respectively.

(Page 2-3, Section 2.3. & Section 2.4., Line 93-101)

Comments 3: Insufficient statistical analysis: The statistical analysis presented needs to be corrected. Studies with two groups and at least two assessment times should be analyzed using a two-way ANOVA with a repeated measures factor (in this case, time). Especially when using GraphPad Prism. Revise the statistical methods to include a two-way repeated measures ANOVA. This approach accounts for within-subject variability and provides a more accurate analysis of the data. Also, what normality test was used?

Response 3: Thank you for pointing this out. We agree with this comment.

A two-tailed unpaired t-test and analysis of variance followed by Dunnett’s multiple comparison test were used to evaluate the differences between the groups.

(Page 3, Section 2.5., Line 127-130)

Comments 4: Text Clarity: The text presented between lines 64 and 66 suggests that 6′-SL may improve health and quality of life. Please review this section and provide a clear rationale for this suggestion as it was not evaluated in the reported study. In addition, ensure that appropriate references are provided to support this claim.

Response 4: Thank you for pointing this out. We agree with this comment.

This section was not accurately evaluated in this study, so the sentence was removed.

Comments 5: [ADDITIONAL WEAKNESSES] Complex sentences and redundancy: Some sentences are overly complex and there are instances of redundant phrases and repetition of ideas. Please simplify complex sentences by breaking them into shorter, more digestible sentences. Eliminate redundant phrases to make the manuscript more concise.

Response 5: Thank you for pointing this out. We agree with this comment.

The English editing was done by Editage, an expert in English editing.

Comments 6: [ADDITIONAL WEAKNESSES] Transitions between sections: Transitions between sections and paragraphs could be smoother. Add transitional phrases or sentences to move the reader more seamlessly through the manuscript. This will improve overall flow and coherence.

Response 6: Thank you for pointing this out. We agree with this comment.

The English editing was done by Editage, an expert in English editing.

Comments 7: [ADDITIONAL WEAKNESSES] Specificity in Methods and Results: Certain sections, especially the Methods and Results, lack specific details and explanations. Please provide more specific details and explanations in these sections to ensure complete clarity and understanding. This will improve the reproducibility of your study.

Response 7: Thank you for pointing this out. We agree with this comment.

We have added more specific explanation in Methods and Results.

Method section:

Page 2, Section 2.1., Line 75-82

Page 2, Section 2.2., Line 86-88

Page 2-3, Section 2.3., Line 93-96

Page 3, Section 2.4., Line 97-101

Page 3, Section 2.5., Line 105-126

Page 3, Section 2.6., Line 128-134

Result section:

Page 5, Section 3.2., Line 155-174

Page 6, Section 3.3, Line 179-190

Comments 8: [MINOR ISSUES] Figures: Revise figures to use the same font as the text. Make sure figures are centered within the text of the manuscript and do not take up the entire page.

Response 8: Thank you for pointing this out. We agree with this comment.

We have revised the font and position of figures.

Comments 9: [Formatting Issues] I. There is an extra parenthesis after the word "biotechnology" on line 109. Please remove the extra parenthesis to correct the formatting.

II. The letters in line 253 are larger than the text. Please adjust the font size to match the rest of the manuscript text.

Response 9: Thank you for pointing this out. We agree with this comment.

We have corrected the formatting and font size.

Comments 10: [Formatting Issues] English language improvements: While the manuscript is generally clear and free of major grammatical errors, further improvements can be made to enhance readability. Make sure that all sentences are well constructed and that the flow of ideas is logical and coherent. Also, continue to use technical terms and concepts correctly and appropriately. Finally, simplify complex sentences, eliminate redundancies, and improve transitions to make the manuscript more accessible and effective for readers.

Addressing these issues, large and small, will greatly improve the rigor, clarity, and overall quality of your manuscript. Thank you for your attention to these details, and I look forward to seeing the improved version of your paper.

Response 10: Thank you for pointing this out. We agree with this comment.

The English editing was done by Editage, an expert in English editing.

4. Response to Comments on the Quality of English Language

Point 1: The manuscript is generally clear and free of major grammatical errors. Sentences are well constructed and the flow of ideas is logical and coherent. Technical terms and concepts are used correctly and appropriately, contributing to the overall readability and comprehension of the manuscript. However, some sentences are overly complex and could be simplified to improve readability. Breaking long sentences into shorter, more digestible ones can help readers follow the argument more easily. In addition, there are instances of redundant phrases and repetition of ideas. Streamlining the text to avoid repetition would make the manuscript more concise.

Response 1: Thank you for pointing this out. We agree with this comment.

The English editing was done by Editage, an expert in English editing.

Point 2: Transitions between sections and paragraphs could be smoother. Adding transitional phrases or sentences would help guide the reader more seamlessly through the manuscript. In some sections, particularly the methods and results, more specific details and explanations would ensure complete clarity and understanding.

Response 2: Thank you for pointing this out. We agree with this comment.

The English editing was done by Editage, an expert in English editing.

We have added transition between paragraphs and transitional phrases.

We have added more specific explanation in Methods and Results as comments 7.

Point 3: Specific suggestions include simplifying complex sentences, eliminating redundancies, and improving transitions. By addressing these areas, you can improve the quality of the English in your manuscript, making it more accessible and effective for your readers.

Response 3: Thank you for pointing this out. We agree with this comment.

The English editing was done by Editage, an expert in English editing

Lastly, I would like to thank you for your time and effort in helping me to improve the quality of this paper. Thank you for your comments, and I hope that my revisions and modifications will meet your expectations.

Reviewer 3 Report

Comments and Suggestions for Authors

Authors suggest that 6’-Sialyllactose has a beneficial effect on muscle fatigue, as evidenced by post-exercise lactate levels. However, the data provided in the current paper are insufficient to conclusively support this conclusion.

1. Please explain how blood BUN (Blood Urea Nitrogen) can be used as a marker of fatigue and how low values are associated with fatigue. Additionally, it might be more appropriate to use blood creatinine concentration as a fatigue marker since high-intensity exercise depletes creatine phosphate (This parameter does not change in your paper).

2. As the treadmill exercise protocol is not described, it is important to specify the intensity (such as speed) of the exercise performed.

3. While the increased expression of OXPHOS complexes protein indicates heightened mitochondrial metabolism, a comprehensive assessment of mitochondria is necessary. Additional experiments should be conducted to evaluate mitochondrial quantity (e.g., citrate synthase activity) and mitochondrial quality (e.g., oxygen consumption measured by a Flux analyzer).

4. To properly assess fatigue tolerance, it would be beneficial to test whether the administration of 6’-Sialyllactose improves endurance exercise capacity, such as through a treadmill running test.

Author Response

For Brief Report

Response to Reviewer 3 Comments

1. Summary

2. Questions for General Evaluation

Reviewer’s Evaluation

Response and Revisions

Does the introduction provide sufficient background and include all relevant references?

Yes

Is the research design appropriate?

Can be improved

Are the methods adequately described?

Must be improved

Are the results clearly presented?

Can be improved

Are the conclusions supported by the results?

Yes

3. Point-by-point response to Comments and Suggestions for Authors

Comments 1: 1. Please explain how blood BUN (Blood Urea Nitrogen) can be used as a marker of fatigue and how low values are associated with fatigue. Additionally, it might be more appropriate to use blood creatinine concentration as a fatigue marker since high-intensity exercise depletes creatine phosphate (This parameter does not change in your paper).

Response 1: Thank you for pointing this out. We agree with this comment.

Please explain how blood BUN (Blood Urea Nitrogen) can be used as a marker of fatigue and how low values are associated with fatigue.

→ It has been known that high BUN level causes kidney disease which include several symptoms such as frequent urination, discolored urine, extreme fatigue, skin itching. Moreover, elimination of BUN has been shown to reverse fatigue.

(Page 8, Section 4, line 234-239)

Therefore, we thought that BUN can be used as a marker of fatigue and low values are associated with anti-fatigue.  

Additionally, it might be more appropriate to use blood creatinine concentration as a fatigue marker since high-intensity exercise depletes creatine phosphate (This parameter does not change in your paper).

→ As you mentioned, it would be better to measure blood creatinine concentration as a fatigue marker in future studies.

Comments 2: 2. As the treadmill exercise protocol is not described, it is important to specify the intensity (such as speed) of the exercise performed.

Response 2: Thank you for pointing this out. We agree with this comment.

We have added more detail explanation of treadmill exercise in Method section as bellows:

2.3. Treadmill Exercise

At 12 weeks after 6′-SL administration, the mice were made to perform treadmill exercise. The mice were first acclimated to a speed of 16.67 m/min for 3 min, and then, the speed was increased by 1.67 m/min every 3 min; the total running time was set to 20 min.

2.4. Blood Lactate and Glucose Measurements

At 12 weeks after 6′-SL administration, blood samples were collected from the tail tip at

the basal level (rest) and 0, 5, and 10 min after the 20-min treadmill exercise session to assess

the blood glucose and lactate levels using a handheld glucometer (ACCU-check, Seoul, South Korea) and a lactate test meter (Lactate Pro 2, Kyoto, Japan), respectively.

(Page 2-3, Section 2.3. & Section 2.4., Line 93-101)

Comments 3: While the increased expression of OXPHOS complexes protein indicates heightened mitochondrial metabolism, a comprehensive assessment of mitochondria is necessary. Additional experiments should be conducted to evaluate mitochondrial quantity (e.g., citrate synthase activity) and mitochondrial quality (e.g., oxygen consumption measured by a Flux analyzer).   

Response 3: Thank you for pointing this out. We agree with this comment.

Further studies to investigate in more detail how 6'-SL is related to lactate clearance in the body would be interesting. For example, changes in glycolysis (glycolysis hexokinase), tricarboxylic acid cycle enzymes (citrate synthase, succinate dehydrogenase, malate dehydrogenase, pyruvate dehydrogenase), oxygen consumption, ATP production, and the Cori cycle may be worth investing.

(Page 9, Section 4, Line 274-277)

Comments 4: To properly assess fatigue tolerance, it would be beneficial to test whether the administration of 6’-Sialyllactose improves endurance exercise capacity, such as through a treadmill running test.

Response 4: Thank you for pointing this out. We agree with this comment.

Further study should be conducted that decreased blood lactate by 6'-SL are related to exercise performance. We have added the sentences in the discussion as follows: 

Moreover, exhausted exercise performance such as treadmill running, swimming, and rotarod test should be performed to confirm whether the reduction in blood lactate level is indeed associated with improved exercise performance in the future.

(Page 8, Section 4, Line 253-256)

Lastly, I would like to thank you for your time and effort in helping me to improve the quality of this paper. Thank you for your comments, and I hope that my revisions and modifications will meet your expectations.

Reviewer 4 Report

Comments and Suggestions for Authors

The title is clear and reflects the study's primary focus.

The discussion interprets the results in the context of existing literature:

The increase in slow MHC and OXPHOS proteins aligns with enhanced oxidative metabolism and reduced muscle fatigue.

The reduction in blood lactate and increase in glucose support improved muscle energy metabolism during exercise.

The study acknowledges the need for further research to confirm these findings in humans.

The discussion also addresses the safety of 6-SL, as no adverse effects were observed on liver and kidney functions or lipid profiles.

The conclusion succinctly summarizes the study's findings and their implications, suggesting that 6-SL is beneficial in reducing muscle fatigue and is safe for potential use in athletic and clinical settings.

The article is well-structured, with a clear hypothesis, detailed methodology, and robust results. The findings are significant and suggest a potential new use for 6-SL in enhancing exercise performance and reducing fatigue. However, the study is limited to animal models, and further research is needed to validate these effects in humans. The study's reliance on a single dosage and duration also suggests areas for further exploration to optimize 6-SL administration.

This article must insist on the fact that it is an ANIMAL study in THE TITLE and that there is nothing to validate the effects of 6-SL on muscle fatigue and exercise performance in HUMAN.

In addition, there is a need to explore different dosages and durations of 6-SL administration and to investigate the underlying mechanisms in greater detail, including molecular pathways involved in lactate metabolism and muscle fiber transformation.

Overall, this is a well-conducted study with promising results that warrant further investigation especially in humans.

Author Response

For Brief Report

Response to Reviewer 4 Comments

1. Summary

2. Questions for General Evaluation

Reviewer’s Evaluation

Response and Revisions

Does the introduction provide sufficient background and include all relevant references?

Can be improved

Is the research design appropriate?

Not applicable

Are the methods adequately described?

Yes

Are the results clearly presented?

Yes

Are the conclusions supported by the results?

Must be improved

3. Point-by-point response to Comments and Suggestions for Authors

Comments 1: The title is clear and reflects the study's primary focus.

Response 1 : We appreciate your comments.

Comments 2: The discussion interprets the results in the context of existing literature.

Response 2 : We appreciate your comments.

Comments 3: The increase in slow MHC and OXPHOS proteins aligns with enhanced oxidative metabolism and reduced muscle fatigue.

Response 3 : We appreciate your comments.

Comments 4: The reduction in blood lactate and increase in glucose support improved muscle energy metabolism during exercise.

Response 4 : We appreciate your comments.

Comments 5: The study acknowledges the need for further research to confirm these findings in humans.

Response 5 : We appreciate your comments.

Comments 6: The discussion also addresses the safety of 6′-SL, as no adverse effects were observed on liver and kidney functions or lipid profiles.

Response 6 : We appreciate your comments.

Comments 7: The conclusion succinctly summarizes the study's findings and their implications, suggesting that 6′-SL is beneficial in reducing muscle fatigue and is safe for potential use in athletic and clinical settings.

Response 7 : We appreciate your comments.

Comments 8: The article is well-structured, with a clear hypothesis, detailed methodology, and robust results. The findings are significant and suggest a potential new use for 6′-SL in enhancing exercise performance and reducing fatigue. However, the study is limited to animal models, and further research is needed to validate these effects in humans. The study's reliance on a single dosage and duration also suggests areas for further exploration to optimize 6′-SL administration.

Response 8 : We appreciate your comments.

We have described the need for further clinical study in conclusion section as follows:

These results suggest that 6'-SL is beneficial and safe in reducing muscle fatigue during exercise, indicating future utility in athletic and clinical settings after further validation.

.  (Page 9, Section 5, line 293-295)

Moreover, we have added the explanation of limitation of this study as follows:

The limitations of this study are that a single dose of 6′-SL (100 mg/kg) was administer for a single duration (12 weeks). It is necessary to optimize the effect of 6′-SL by conducting experiments under various conditions, such as the concentration at which there is no effect, concentration at which there is the best effect, and minimum period for which the effect appears in the future.

(Page 9, Section 4, line 283-287)

Comments 9: This article must insist on the fact that it is an ANIMAL study in THE TITLE and that there is nothing to validate the effects of 6′-SL on muscle fatigue and exercise performance in HUMAN.

Response 9 : We appreciate your comments.

To insist on the fact that it is an animal study, we have changed the title “6´-Sialyllactose suppresses muscle fatigue through reduced blood lactate” to ““6′-Sialyllactose Alleviates Muscle Fatigue through Reduced Blood Lactate Level after Treadmill Exercise in Mice”.

(Page 1, Title, Line 2-3)

Comments 10: In addition, there is a need to explore different dosages and durations of 6′-SL administration and to investigate the underlying mechanisms in greater detail, including molecular pathways involved in lactate metabolism and muscle fiber transformation.

Response 10 : We appreciate your comments.

We have added the limitations as a future study in discussion section as follows:

The limitations of this study are that a single dose of 6′-SL (100 mg/kg) was administer for a single duration (12 weeks). It is necessary to optimize the effect of 6′-SL by conducting experiments under various conditions, such as the concentration at which there is no effect, concentration at which there is the best effect, and minimum period for which the effect appears in the future.

(Page 9, Section 4, line 283-287)

Further studies to investigate how 6'-SL is related to lactate clearance in the body would be interesting. For example, changes in glycolysis (glycolysis hexokinase), tricarboxylic acid cycle enzymes (citrate synthase, succinate dehydrogenase, malate dehydrogenase, pyruvate dehydrogenase), oxygen consumption, ATP production, and the Cori cycle may be worth investing.

Measuring the level of lactate in muscles and the liver will allow more precise explanation of the mechanism by which blood lactate level is reduced. The expression of each muscle fiber type (type I, type IIa, type IIb, and type IIx) needs to be examined using western blotting and immunofluorescence to examine muscle fiber switching during 6'-SL administration. (Page 9, Section 4, Line 273-282)s

Comments 11: Overall, this is a well-conducted study with promising results that warrant further investigation especially in humans.

Response 11: We appreciate your comments.

Lastly, I would like to thank you for your time and effort in helping me to improve the quality of this paper. Thank you for your comments, and I hope that my revisions and modifications will meet your expectations.

Round 2

Reviewer 1 Report

Comments and Suggestions for Authors

Thanks for response to the comments and did some revisions.

(1) Line 76, Line 80, ??

(2) Line 82, how to produce the 6'-SL by the enzyme synthesis method? 

(3) Where are the FIGs of AUC in the Line 165, Line 169?

(4) In Figure 3, what are the Molecular weight of proteins——Slow MHC, Fast MHC, tubulin? They are lack in the WB bands

Author Response

For Brief Report

Response to Reviewer 1 Comments (Round 2)

1. Summary

2. Questions for General Evaluation

Reviewer’s Evaluation

Response and Revisions

Does the introduction provide sufficient background and include all relevant references?

Can be improved

Is the research design appropriate?

Can be improved

Are the methods adequately described?

Yes

Are the results clearly presented?

Can be improved

Are the conclusions supported by the results?

Can be improved

3. Point-by-point response to Comments and Suggestions for Authors

Comments 1: (1) Line 76, Line 80, ??

Response 1: Thank you for pointing this out. 

→ There was an incorrect description, so we have amended it as follows:

Animal health was monitored by animal activity, panting, fur condition, body weight, body weight gain, and food intake. Animal activity, panting, and fur condition were monitored once daily, 5 days per week. Body weight, body weight gain, and food intake were measured weekly.

(Page 2, Section 2.1, Line 76-79)

Comments 2: (2) Line 82, how to produce the 6'-SL by the enzyme synthesis method? 

Response 2: Thank you for pointing this out.

→ We have added the explanations how to produce 6'-SL as follows:

6′-SL was produced using the one pot reaction system invented at GeneChem, Inc. (Daejeon, South Korea) and dissolved in water. Briefly, substrates and enzymes (cytidylate kinase (CMK), acetate kinase (ACK), CMP-NeuAc synthetase (NEU), N-acetyl-D-glucosamine-2-epimerase (NANE), NeuAc aldolase (NAN), and α2,6-sialyltranferase (α2,6STN)) were mixed and reacted in one-pot reactor. These enzymes were obtained from a strain of beta-D-galactosidase deficient E. coli BW25113 that originated from non-pathogenic E. coli K-12.

(Page 2, Section 2.1, Line 86-92)

Comments 3: (3) Where are the FIGs of AUC in the Line 165, Line 169?

Response 3: Thank you for pointing this out.

→ We have added the figures of AUC in Figure 2.

(Page 6, Section 3.2, Figure 2, Line 178-180, 183-184)

Comments 4: (4) In Figure 3, what are the Molecular weight of proteins——Slow MHC, Fast MHC, tubulin? They are lack in the WB bands.

Response 4: Thank you for pointing this out.

→ We have added the molecular weight of proteins.

(Page 7, Section 3.2, Figure 3)

Lastly, I would like to thank you for your time and effort in helping me to improve the quality of this paper. Thank you for your comments, and I hope that my revisions and modifications will meet your expectations.

Reviewer 2 Report

Comments and Suggestions for Authors

Dear Authors,

I have reviewed the revised version of your manuscript entitled "6′-sialyllactose suppresses muscle fatigue through reduced blood lactate". The second version of the manuscript, ID nutrients-3121908, has successfully addressed most of the comments from the first version, especially regarding details on the activity regime of the mice, treadmill exercise protocol, text clarity, complex sentences, transitions, specificity of methods and results, figures, formatting issues, and English language improvements.

I appreciate the significant improvements you have made in response to the previous comments. However, the statistical analysis section still does not include the recommended two-way repeated measures ANOVA. Since the new techniques used do not reduce statistical precision, I agree with only partial adoption of the recommended tests.

Below is a brief presentation of the points that require further attention:
1. Consistency in Terminology: Ensure that terminology is used consistently throughout the manuscript. For example, terms like "6′-sialyllactose," "6′-SL," and "sialyllactose" should be uniformly used. My recommendation is to standardize the terminology used to avoid confusion.
2. Expanded Discussion on Mechanisms: While the discussion covers the main findings, there could be more depth in explaining the underlying mechanisms of how 6′-SL affects muscle metabolism. My recommendation is to expand on the discussion of potential biochemical and molecular pathways involved in 6′-SL's effects on muscle fiber types and oxidative phosphorylation complexes.
3. Explore the possibility of providing specific details about the instruments and protocols used for data collection, including any calibration procedures or settings for measuring instruments.

In conclusion, you have made significant progress in addressing the previous comments and the manuscript has improved significantly. However, consider these points for further revisions to improve the quality of the manuscript.

Comments on the Quality of English Language

Double-check for any remaining typographical errors or inconsistencies in formatting.

Ensure that all abbreviations used in the manuscript are listed and defined at their first occurrence.

Author Response

For Brief Report

Response to Reviewer 2 Comments (Round 2)

1. Summary

2. Questions for General Evaluation

Reviewer’s Evaluation

Response and Revisions

Does the introduction provide sufficient background and include all relevant references?

Yes

Is the research design appropriate?

Yes

Are the methods adequately described?

Yes

Are the results clearly presented?

Yes

Are the conclusions supported by the results?

Yes

3. Point-by-point response to Comments and Suggestions for Authors

I have reviewed the revised version of your manuscript entitled "6′-sialyllactose suppresses muscle fatigue through reduced blood lactate". The second version of the manuscript, ID nutrients-3121908, has successfully addressed most of the comments from the first version, especially regarding details on the activity regime of the mice, treadmill exercise protocol, text clarity, complex sentences, transitions, specificity of methods and results, figures, formatting issues, and English language improvements.

Comments 1: I appreciate the significant improvements you have made in response to the previous comments. However, the statistical analysis section still does not include the recommended two-way repeated measures ANOVA. Since the new techniques used do not reduce statistical precision, I agree with only partial adoption of the recommended tests.

Response 1: Thank you for pointing this out. We agree with this comment.

→ We have included the two-way analysis of variance followed by Tukey’s multiple comparison test in Figure 1 and Figure 2.

(Page 4-5, Section 3.1, Line 160)

(Page 6, Section 3.2, Line 192)

Comments 2:  Below is a brief presentation of the points that require further attention:

Response 2: Thank you for pointing this out. We agree with this comment.

1. Consistency in Terminology: Ensure that terminology is used consistently throughout the manuscript. For example, terms like "6′-sialyllactose," "6′-SL," and "sialyllactose" should be uniformly used. My recommendation is to standardize the terminology used to avoid confusion.

→ We have unified the terms 6′-Sialyllactose (6′-SL). In its first appearance, 6′-sialyllactose was used, and from the second appearance onwards, 6′-SL was used.

→ We have unified the terms serum lactate or glucose to blood lactate or glucose.

2. Expanded Discussion on Mechanisms: While the discussion covers the main findings, there could be more depth in explaining the underlying mechanisms of how 6′-SL affects muscle metabolism. My recommendation is to expand on the discussion of potential biochemical and molecular pathways involved in 6′-SL's effects on muscle fiber types and oxidative phosphorylation complexes.

→ We have expanded the discussion on mechanism as follows:

[muscle fiber types]

These reports support that 6′-SL-indcued protein expression of slow MHC will enhance anti-fatigue in muscle (Figure 3A). However, the precise mechanism of 6'-SL-induced slow-MHC protein expression is not clear yet. It will be necessary to investigate the muscle fiber switching through evaluating the expression of each muscle fiber type (type I, type IIa, type IIb, and type IIx) using western blotting and immunofluorescence, as well as the expression of myogenic regulatory factors, myostatin, paired box family, and forkhead box transcription factors, which are known to control myofiber development and differentiation [30].

(Page 8, Section 4, Line 236-243)

[OXPHOS complexes]

6′-SL increased protein expression of OXPHOS complexes in the GAS muscle in our study (Figure 3B). Further study will be needed to evaluate molecular mechanisms involved in effects of 6'-SL on OXPHOS complex proteins. The relevance of protein kinases such as cAMP-dependent protein kinase A, PTEN-induced putative kinase 1, cyclin-dependent kinase 11, SRC, and 5′-AMP-activated protein kinase, which are known to regulate the OXPHOS system, could be improve the quality of this study [35].

(Page 8, Section 4, Line 253-258)

[lactate metabolism]

In our study, 6′-SL decreased blood lactate level and increased blood glucose level after the treadmill exercise (Figure 2). Therefore, we speculate that 6′-SL may enhances exercise performance by decreasing lactate and BUN levels and enhancing glucose supply in the blood. Further studies to investigate how 6'-SL regulate lactate metabolism during exercise would be interesting. For example, changes in glycolysis (glycolysis hexokinase), tricarboxylic acid cycle enzymes (citrate synthase, succinate dehydrogenase, malate dehydrogenase, pyruvate dehydrogenase), oxygen consumption, and ATP production may be worth investing. Moreover, measuring the level of lactate in muscle and livers, and assessing the relevance of Cori cycle will more precise explanation of how blood lactate levels decline.

(Page 8, Section 4, Line 270-279)

3. Explore the possibility of providing specific details about the instruments and protocols used for data collection, including any calibration procedures or settings for measuring instruments.
In conclusion, you have made significant progress in addressing the previous comments and the manuscript has improved significantly. However, consider these points for further revisions to improve the quality of the manuscript.

→ We have provided details in Materials and Methods section.

(Page 2, Section 2.1, Line 71-93)

(Page 3, Section 2.3, Line 104)

(Page 3, Section 2.6, Line 138-147)

4. Comments on the Quality of English Language

Double-check for any remaining typographical errors or inconsistencies in formatting.

Ensure that all abbreviations used in the manuscript are listed and defined at their first occurrence.

Response 1: Thank you for pointing this out.

→ We have checked abbreviation and errors in the manuscript.

Lastly, I would like to thank you for your time and effort in helping me to improve the quality of this paper. Thank you for your comments, and I hope that my revisions and modifications will meet your expectations.

Reviewer 3 Report

Comments and Suggestions for Authors

Although the description of methods has been enhanced, no additional data have been presented, resulting in a lack of scientific basis for the conclusions. Therefore, further improvements are necessary for publication.

Author Response

For Brief Report

Response to Reviewer 3 Comments (Round 2)

1. Summary

2. Questions for General Evaluation

Reviewer’s Evaluation

Response and Revisions

Does the introduction provide sufficient background and include all relevant references?

Yes

Is the research design appropriate?

Must be improved

Are the methods adequately described?

Yes

Are the results clearly presented?

Can be improved

Are the conclusions supported by the results?

Must be improved

3. Point-by-point response to Comments and Suggestions for Authors

Comments 1: Although the description of methods has been enhanced, no additional data have been presented, resulting in a lack of scientific basis for the conclusions. Therefore, further improvements are necessary for publication.

Response 1: Thank you for pointing this out. We agree with this comment.

→ Unfortunately, results you mentioned are lacking in this study and therefore, we have decided to submit our study to Brief Report. However, we will improve the results according to your comments in the future. Therefore, we have mentioned that further study will be needed to clarify the detailed mechanism of 6´-SL’s effects on muscle fiber types, oxidative phosphorylation complexes, and lactate metabolism as follows:

[muscle fiber types]

These reports support that 6′-SL-indcued protein expression of slow MHC will enhance anti-fatigue in muscle (Figure 3A). However, the precise mechanism of 6'-SL-induced slow-MHC protein expression is not clear yet. It will be necessary to investigate the muscle fiber switching through evaluating the expression of each muscle fiber type (type I, type IIa, type IIb, and type IIx) using western blotting and immunofluorescence, as well as the expression of myogenic regulatory factors, myostatin, paired box family, and forkhead box transcription factors, which are known to control myofiber development and differentiation [30].

(Page 8, Section 4, Line 236-243)

[OXPHOS complexes]

6′-SL increased protein expression of OXPHOS complexes in the GAS muscle in our study (Figure 3B). Further study will be needed to evaluate molecular mechanisms involved in effects of 6'-SL on OXPHOS complex proteins. The relevance of protein kinases such as cAMP-dependent protein kinase A, PTEN-induced putative kinase 1, cyclin-dependent kinase 11, SRC, and 5′-AMP-activated protein kinase, which are known to regulate the OXPHOS system, could be improve the quality of this study [35].

(Page 8, Section 4, Line 253-258)

[lactate metabolism]

In our study, 6′-SL decreased blood lactate level and increased blood glucose level after the treadmill exercise (Figure 2). Therefore, we speculate that 6′-SL may enhances exercise performance by decreasing lactate and BUN levels and enhancing glucose supply in the blood. Further studies to investigate how 6'-SL regulate lactate metabolism during exercise would be interesting. For example, changes in glycolysis (glycolysis hexokinase), tricarboxylic acid cycle enzymes (citrate synthase, succinate dehydrogenase, malate dehydrogenase, pyruvate dehydrogenase), oxygen consumption, and ATP production may be worth investing. Moreover, measuring the level of lactate in muscle and livers, and assessing the relevance of Cori cycle will more precise explanation of how blood lactate levels decline.

(Page 8, Section 4, Line 270-279)

Lastly, I would like to thank you for your time and effort in helping me to improve the quality of this paper. Thank you for your comments, and I hope that my revisions and modifications will meet your expectations.

Reviewer 4 Report

Comments and Suggestions for Authors

This article presents a study on the effects of 6′-Sialyllactose (6′-SL), a human milk oligosaccharide (HMO), on exercise performance and muscle fatigue in mice. Here is a review of the article, assessing its structure, methodology, results, and overall conclusions:

This manuscript has been improved.  The background information is comprehensive, and the hypothesis—that 6′-SL may improve muscle fatigue during exercise by influencing muscle characteristics and mitochondrial function—is clearly stated.

   - The study shows that 6′-SL reduces blood lactate levels and increases glucose levels after exercise, suggesting improved energy metabolism. The findings that 6′-SL enhances slow MHC protein expression and increases OXPHOS complex proteins in muscle tissue support its potential role in promoting oxidative metabolism and reducing fatigue.

   - The increase in slow MHC and OXPHOS complex expression suggests a shift towards oxidative metabolism, which aligns with the observed decrease in lactate production. This is significant as it suggests a mechanistic basis for the anti-fatigue effects of 6′-SL.

-The discussion effectively interprets the results, linking increased slow-twitch muscle fiber expression to enhanced fatigue resistance and improved exercise performance. The article places the findings in the context of existing literature, highlighting the novelty of using 6′-SL as an exercise performance enhancer. However, the discussion could benefit from a deeper exploration of the potential molecular pathways involved, as well as comparisons with other known supplements or interventions that affect lactate metabolism and mitochondrial function.

-The article acknowledges several limitations, including the use of a single dose and duration for 6′-SL administration. Exploring a range of doses and treatment durations would help determine the optimal conditions for its effects. The study also lacks direct measurements of muscle fiber type switching and intracellular metabolic changes, which could provide more insight into the mechanisms of action.

The conclusions are well-supported by the data, but further studies, particularly in humans, are needed to validate these findings and explore their clinical relevance: you have to insist on that point.

Indeed, the article presents a well-structured and comprehensive study that contributes valuable insights into the potential use of 6′-SL for improving exercise performance. The methodology is sound, and the results are clearly presented and analyzed.

However, future research should address the study's limitations, including dose optimization, mechanistic pathways, and the translation of these findings to human subjects. Overall, the study is a promising step towards understanding the role of 6′-SL in muscle physiology and exercise performance.

Author Response

For Brief Report

Response to Reviewer 4 Comments (Round 2)

1. Summary

2. Questions for General Evaluation

Reviewer’s Evaluation

Response and Revisions

Does the introduction provide sufficient background and include all relevant references?

Yes

Is the research design appropriate?

Yes

Are the methods adequately described?

Yes

Are the results clearly presented?

Yes

Are the conclusions supported by the results?

Must be improved

3. Point-by-point response to Comments and Suggestions for Authors

This article presents a study on the effects of 6′-Sialyllactose (6′-SL), a human milk oligosaccharide (HMO), on exercise performance and muscle fatigue in mice. Here is a review of the article, assessing its structure, methodology, results, and overall conclusions:

This manuscript has been improved.  The background information is comprehensive, and the hypothesis—that 6′-SL may improve muscle fatigue during exercise by influencing muscle characteristics and mitochondrial function—is clearly stated.

 - The study shows that 6′-SL reduces blood lactate levels and increases glucose levels after exercise, suggesting improved energy metabolism. The findings that 6′-SL enhances slow MHC protein expression and increases OXPHOS complex proteins in muscle tissue support its potential role in promoting oxidative metabolism and reducing fatigue.

  - The increase in slow MHC and OXPHOS complex expression suggests a shift towards oxidative metabolism, which aligns with the observed decrease in lactate production. This is significant as it suggests a mechanistic basis for the anti-fatigue effects of 6′-SL.

-The discussion effectively interprets the results, linking increased slow-twitch muscle fiber expression to enhanced fatigue resistance and improved exercise performance. The article places the findings in the context of existing literature, highlighting the novelty of using 6′-SL as an exercise performance enhancer. 

However, the discussion could benefit from a deeper exploration of the potential molecular pathways involved, as well as comparisons with other known supplements or interventions that affect lactate metabolism and mitochondrial function.

Response: Thank you for pointing this out. We agree with this comment.

→ we have mentioned that further study will be needed to clarify the detailed mechanism of 6´-SL’s effects on mitochondria function (oxidative phosphorylation complexes) and lactate metabolism as follows:

[OXPHOS complexes]

6′-SL increased protein expression of OXPHOS complexes in the GAS muscle in our study (Figure 3B). Further study will be needed to evaluate molecular mechanisms involved in effects of 6'-SL on OXPHOS complex proteins. The relevance of protein kinases such as cAMP-dependent protein kinase A, PTEN-induced putative kinase 1, cyclin-dependent kinase 11, SRC, and 5′-AMP-activated protein kinase, which are known to regulate the OXPHOS system, could be improve the quality of this study [35].

(Page 8, Section 4, Line 253-258)

[lactate metabolism]

In our study, 6′-SL decreased blood lactate level and increased blood glucose level after the treadmill exercise (Figure 2). Therefore, we speculate that 6′-SL may enhances exercise performance by decreasing lactate and BUN levels and enhancing glucose supply in the blood. Further studies to investigate how 6'-SL regulate lactate metabolism during exercise would be interesting. For example, changes in glycolysis (glycolysis hexokinase), tricarboxylic acid cycle enzymes (citrate synthase, succinate dehydrogenase, malate dehydrogenase, pyruvate dehydrogenase), oxygen consumption, and ATP production may be worth investing. Moreover, measuring the level of lactate in muscle and livers, and assessing the relevance of Cori cycle will more precise explanation of how blood lactate levels decline.

(Page 8, Section 4, Line 270-279)

→ One of our study limitations, we have added the following comment about its benefits compared to other supplements:

Third, the efficacy of 6'-SL in reducing muscle fatigue is more robust when compared to other nutritional supplements known to be effective in removing lactate from the blood, such as omega-3, CoQ10, magnesium, and acetyl-L-carnitine in the future. To support these findings and develop 6'-SL as a dietary supplement to alleviate muscle fatigue, exploring clinical relevance will be required.

(Page 9, Section 4, Line 308-312)

-The article acknowledges several limitations, including the use of a single dose and duration for 6′-SL administration. Exploring a range of doses and treatment durations would help determine the optimal conditions for its effects. The study also lacks direct measurements of muscle fiber type switching and intracellular metabolic changes, which could provide more insight into the mechanisms of action.

Response: Thank you for pointing this out. We agree with this comment.

→ we have mentioned that further study will be needed to clarify the detailed mechanism of 6´-SL’s effects on muscle fiber types and its mechanism as follows:

These reports support that 6′-SL-indcued protein expression of slow MHC will enhance anti-fatigue in muscle (Figure 3A). However, the precise mechanism of 6'-SL-induced slow-MHC protein expression is not clear yet. It will be necessary to investigate the muscle fiber switching through evaluating the expression of each muscle fiber type (type I, type IIa, type IIb, and type IIx) using western blotting and immunofluorescence, as well as the expression of myogenic regulatory factors, myostatin, paired box family, and forkhead box transcription factors, which are known to control myofiber development and differentiation [30].

(Page 8, Section 4, Line 236-243)

The conclusions are well-supported by the data, but further studies, particularly in humans, are needed to validate these findings and explore their clinical relevance: you have to insist on that point.

Response: Thank you for pointing this out. We agree with this comment.

→ We have added the explanations about further studies on humans as follows:

To support these findings and develop 6'-SL as a dietary supplement to alleviate muscle fatigue, exploring clinical relevance will be required.

(Page 9, Section 4, Line 310-312)

Indeed, the article presents a well-structured and comprehensive study that contributes valuable insights into the potential use of 6′-SL for improving exercise performance. The methodology is sound, and the results are clearly presented and analyzed.

However, future research should address the study's limitations, including dose optimization, mechanistic pathways, and the translation of these findings to human subjects. Overall, the study is a promising step towards understanding the role of 6′-SL in muscle physiology and exercise performance.

Response: Thank you for your comments.

→ We have added our study limitations as follows:

This study has several limitations. First, a single dose of 6′-SL (100 mg/kg) was administer for a single duration (12 weeks) in this study. It is necessary to optimize the effect of 6′-SL by conducting experiments under various conditions, such as the concentration at which there is no effect, concentration at which there is the best effect, and minimum period for which the effect appears in the future. Second, only one type of exercise was tested to measure blood lactate and glucose levels. Treadmill running is physically demanding and is an effective exercise to burn energy and induce fatigue; therefore, this type of exercise was chosen in this study to measure blood lactate and blood glucose levels after inducing fatigue. However, in the future, another type of exercise, such as swimming, should be conducted to obtain more detailed results on the effectiveness of 6'-SL in reducing lactate levels. Moreover, exhausted exercise performance such as treadmill running, swimming, and rotarod test should be performed to confirm whether the reduction in blood lactate level is indeed associated with improved exercise performance in the future. Third, the efficacy of 6'-SL in reducing muscle fatigue is more robust when compared to other nutritional supplements known to be effective in removing lactate from the blood, such as omega-3, CoQ10, magnesium, and acetyl-L-carnitine in the future. To support these findings and develop 6'-SL as a dietary supplement to alleviate muscle fatigue, exploring clinical relevance will be required.   

(Page 9, Section 4, Line 295-312)

Lastly, I would like to thank you for your time and effort in helping me to improve the quality of this paper. Thank you for your comments, and I hope that my revisions and modifications will meet your expectations.

Round 3

Reviewer 3 Report

Comments and Suggestions for Authors

Thank you for answering my review point. I do not have additional comments